# Applicability of the Low-Cost Optical Particle Counter OPC-N3 for Microphysical Measurements of Fog

Katarzyna Nurowska[1], Moein Mohammadi[1], Szymon Malinowski[1], and Krzysztof Markowicz[1]

[1]Institute of Geophysics, Faculty of Physics, University of Warsaw, Warsaw, Poland

**Correspondence:** Katarzyna Nurowska (knurowska@fuw.edu.pl)

**Abstract.** Low-cost devices for particulate matter measurements are characterized by small dimensions and a light weight. This advantage makes them ideal for UAV measurements, where those parameters are crucial. However, they also have some issues. The values of particulate matter from low-cost optical particle counters can be biased by high ambient humidity. In this article, we evaluate low-cost optical particle counter Alphasense OPC-N3 for measuring the microphysical properties of fog. This study aimed to show that OPC-N3 not only registers aerosols or humidified aerosols but also registers fog droplets.

The study was carried out on the rooftop of the Institute of Geophysics, University of Warsaw, Poland, during autumn-winter 2021. To validate the results, the data from OPC-N3 were compared with the data obtained from the reference instrument, the Oxford Laser VisiSize D30. VisiSize D30 is a shadowgraph device able to register photos of individual droplets.

Considering the effective radius of droplets, it is possible to differentiate low-visibility situations between fog conditions (which are not hazardous for people) from haze events, when highly polluted air can cause health risks to people.

The compared microphysical properties were liquid water content (LWC), number concentration ($N_c$), effective radius $r_{eff}$ and statistical moments of radius. The Pearson correlation coefficient between both devices for LWC was 0.92, $N_c$ was 0.95, and for $r_{eff}$ was 0.63. Overall, these results suggest good compliance between instruments. However, the OPC-N3 has to be corrected regarding professional equipment.

## 1 Introduction

Fog is defined as a layer of water droplets (or ice crystals) near the Earth's surface, which reduces visibility to below 1 km (Gultepe, 2007). Depending on the cause which helps in forming it, there are different types of fog, e.g., radiation fog, advection fog, and valley fog. Radiation fog forms in the evening when the heat absorbed by the Earth's surface during the day is radiated into the air. The air near the ground cools down when the air reaches saturation, and the first layer of fog forms. Fog phenomenon can affect from the small scale (valley fog) to region of the whole country.

Fog is a dangerous phenomenon for transport safety (Gultepe et al., 2015; Bartok et al., 2012), which causes increased road accidents as well as downtime of aircraft flights. This phenomenon affects the carriage of goods by land, air and maritime transport, causing financial losses similar to those caused by extreme weather events (Kulkarni et al., 2019; Price et al., 2018).

The research by Vautard et al. (2009) shows that fog in Europe has decreased significantly in the last 30 years. This can be the outcome of an increase in air temperature (Klemm and Lin (2016)). However, the fog decrease trend is spatially and

temporally correlated with a negative trend in sulphur dioxide emissions, indicating that air quality policies significantly impact fog occurrence (Vautard et al., 2009).

Particulate matter (PM) is a suspension of solid or liquid particles in the air, such as, for example, dust or soot. Aerosol is a critical component that contributes to fog formation. A decrease in PM can cause changes in fog occurrence or its microphysics. Fog is generated with relatively low super-saturation (SS). Mazoyer et al. (2022) showed that the amount of activated aerosol depends more on the aerosol size than its chemical composition. Numerical simulations conducted by Tsai et al. (2021) show that for urban aerosols, a higher density of aerosols resulted in more but smaller droplets and increased liquid water content (LWC). The effect of urban aerosols was more pronounced near the surface, where the aerosols were most concentrated.

The decrease in fog events in Europe coincides with the reduction of particulate matter. Between 1998 and 2010, Barmpadimos et al. (2012) found that in Europe, the concentration of $PM_{10}$ (particulate matter of sizes less than 10 μm) decreased, the mean trends were -0.4 $\mu g\,m^{-3}$ per year. According to Colette and Rouïl (2021) the median annual concentration of $PM_{10}$ decreased by 40 % in Europe from 2000 to 2017. The decrease in $PM_{2.5}$ after 2008 was visible, but less pronounced. PM concentrations decrease due to the reduction of primary PM emissions, but also due to the reduction of PM precursors, such as $SO_x$, $NO_x$ and $NH_3$. The mentioned studies agree with Beloconi and Vounatsou (2021) where it was shown that over a span of 14 years (2006-2019), the concentrations of $PM_{10}$ and $PM_{2.5}$ decreased by 36.5% and 39.1%, respectively.

The decrease in $PM_{10}$ and $PM_{2.5}$ is thanks to social awareness. Many countries have introduced restrictions on the amount of $PM_{10}$ and $PM_{2.5}$ emitted during the day and year. Thanks to the growing awareness of the community, the networks measuring the concentration of $PM_{10}$ and $PM_{2.5}$ are growing (Considine et al., 2021; Feinberg et al., 2019). Consequently,increasingly inexpensive and simple PM sensors are available on the market.

Low-cost optical particle counters (OPC) are based on measuring light scattering on suspended particles in the air. Light scattering can be enhanced by water droplets. Thus, the reported values of PM are overestimated because of the water uptake by the particles. It is shown that for some low-cost OPCs, the measured PM is dependent on ambient humidity (e.g., Liu et al., 2019; Wang et al., 2015; Badura et al., 2018; Jayaratne et al., 2018). To remove the bias, the air should be dehumidified or correction methods should be applied, as proposed by Crilley et al. (2018).

Most fog studies focus on ground measurements or numerical modeling. Current weather forecasts have a problem with the high precision of fog forecasting. This is due to many processes in the lower boundary layer that are involved in its evolution. The processes that influence the development and duration of fog are, for example, droplet microphysics, aerosol hygroscopicity or radiation and turbulent processes (Gultepe et al., 2007). The local surface fluctuations of temperature or humidity have a high impact on the model prediction of fog conditions (Hu et al., 2017).

Microphysical models are based on microphysical parameters such as the total droplet number concentration $N_c$, mean volume droplet diameter $\overline{D_V}$, LWC (Gultepe et al., 2017). Based on in situ measurements, many works provide the range for LWC: $0.01 - 0.5\,\mathrm{g\,m^{-3}}$ (although the value typically does not exceed $0.2\,\mathrm{g\,m^{-3}}$); $N_c$: $10 - 500$ cm$^{-3}$ and $\overline{D}$: $10 - 20$ μm (e.g., Pruppacher and Klett, 2010; Liu et al., 2011; Tsai et al., 2021; Mazoyer et al., 2019, 2022). However, these parameters show a large variability.

The Fog Monitor FM-120 from Droplet Measurement Technologies is the standard device for measuring fog microphysics. It is an optical spectrometer that can measure the droplet size distribution ($DSD(r)$) from 2 μm to 50 μm, FM-120 can generate particle-by-particle files. It can be used to obtain information on $N_c$, LWC, and visibility. This instrument is widely used to measure fog, and allows for consistent comparison based on various field observations (e.g., Weston et al., 2022; Gultepe et al., 2021; Mazoyer et al., 2019; Degefie et al., 2015; Gultepe et al., 2009).

Another in situ device is Cloud Droplet Probe 2 (CDP-2), which illuminates the droplets and measures the intensity of the forward-scattered light. Droplets are measured in the range of 2 μm to 50 μm and $N_c$ to 2000 cm$^{-3}$. This device was used, among other things, during two campaigns oriented towards fog microphysics: High Energy Laser in Fog project (HELFOG) and the Toward Improving Coastal Fog Prediction project (C-FOG) (Wang et al., 2021; Gultepe et al., 2021).

The alternative device used to retrieve the aerosol and fog particle size distribution is the Palas Welas-2000 particle counter. The Palas Welas-2000 is a system capable of the analysis of light scattering by a single particle. The device can measure particles in four ranges: $0.2 - 10$ μm, $0.3 - 17$ μm, $0.6 - 40$ μm or $2 - 100$ μm. The Palas Welas-2000 spectrometer was used in the ParisFog fog campaign (Haeffelin et al., 2010). ParisFog lasted 6 months between October 2006 and March 2007. The measurements were taken at the SIRTA observatory 20 km south of Paris and a total of 154 hours of fog were recorded. The articles Elias et al. (2009a) and Elias et al. (2009b) focus on the amount of extinction caused by each particle mode in clear sky, haze (pre-fog) conditions and after fog onset.

Optical devices, such as the lidar or the celiometer, cannot penetrate thick fog to retrieve information about their vertical structure (Okamoto et al., 2016; Xu et al., 2022). On the contrary, radars based on micrometer wavelengths can penetrate the fog and measure the reflectivity and velocity structure (Hamazu et al., 2003). The HATPRO microwave radiometer (MWR) registers a vertical brightness temperature where the highest resolution (200 m) is in the range of 0 to 2 km. Based on several brightness channels, the HATPRO artificial neural network algorithm inverts the radiance to estimate the vertical profiles of atmospheric temperature ($T$), relative humidity (RH), integrated water vapor, liquid water path (LWP) and liquid water content (LWC) vertical profile. SOFOG3D (SOuth west FOGs 3D experiment) was a project that was conducted from October 2019 to April 2020 in France over 300×300 km, where eight MWRs were used. The data from this campaign have been presented at scientific conferences (e.g., Martinet et al., 2021; Vishwakarma et al., 2021; Burnet et al., 2020). The reason for the use of MWR during the SOFOG3D campaign was the article Martinet et al. (2020), which showed the improvement of numerical weather prediction models by the assimilation of MWR data.

The microphysical properties of fog are not the only ones that influence the life cycle of fog. Radiative heating and cooling are crucial to the development and decay of radiation fog. Wærsted et al. (2017) showed that for fog that has no cloud above and its liquid water path exceeds $30$ g m$^{-2}$, the cooling of the fog by long-wave radiation can produce $40 - 70$ g m$^{-2}$ h$^{-1}$ liquid water by condensation. Furthermore, the heating of short-wave radiation can contribute significantly to the evaporation by $10 - 15$ g m$^{-2}$ h$^{-1}$ in thick fog in the middle of the day (winter).

The motivation for this work is a lack of devices that can perform in situ microphysical measurements of the vertical structure of the fog. For this purpose, miniature devices are needed that can be mounted on the Unnamed Aircraft Vehicle

(UAV), balloons or cable cars. In this study, we propose a novel application of Alphasense OPC-N3 optical particulate matter. We postulate that OPC-N3 can be used to observe the microphysical properties of fog.

During this study, we observed fog near the city center of Warsaw at the Institute of Geophysics, University of Warsaw. Parameters such as LWC, $N_c$ and the effective radius ($r_{eff}$) are examined. The results from Alphasense OPC-N3 were validated using the Oxford Laser VisiSize D30 (ShadowGraph instrument), which has already been successfully used to study cloud/fog microphysics (Nowak et al., 2021; Mohammadi et al., 2022).

The rest of this article is organized as follows: Section 2 describes the data acquisition and the apparatus used, Section 3 describes data processing. Section 4 shows the results of the comparison between OPC-N3 and ShadowGraph. Section 5 reports a case study of fog evolution during night observation. Section 6 discusses the applicability of OPC-N3 to perform fog microphysics measurements.

## 2 Instruments

### 2.1 Data acquisition

The measurements were conducted at the Institute of Geophysics (52.21°N, 20.98°E, 115 m a.s.l.), Faculty of Physics, University of Warsaw, Poland. The Radiation Transfer Laboratory is located on the rooftop of the Faculty of Physics, where measurements of the optical and microphysical properties of atmospheric aerosols and clouds are continuously conducted (i.e., by nephelometer Ecotech Aurora4000), as well as components of radiation fluxes and sensible and latent heat fluxes at the Earth's surface (CGR4 for incoming long-wave (infrared) flux, CMP22 total incoming short-wave (solar) flux). Basic atmospheric parameters are collected by Vaisala weather station WXT 520. The platform is located 20 m above the ground. Some devices, such as Oxford Laser VisiSize D30, are mounted solely for a given period of time.

The fog observations were carried out in the autumn - winter period of 2021. As ShadowGraph has a waterproof case, it was mounted on the platform for the duration of the observation. The OPC-N3 was mounted next to it in case of high probability of fog events without any protection. The Figure1 presents the mounting of both devices at The Radiation Transfer Laboratory. A total of five days with fog were recorded, most of the fog occurred at night, and Table 1 summarises these events. During D-16.11, the most extended period of fog occurred and was analyzed in detail in Section 5.

**Table 1.** General information of the measurement cases analyzed in this study, including the date, time and length of measurements.

| Referred as | Measurement start [UTC] | Measurement stop [UTC] | Day / Night | Total minutes of fog |
|---|---|---|---|---|
| D-12.11 | 12.11.2021 19:11 | 13.11.2021 04:43 | Night | 150 |
| D-16.11 | 16.11.2021 22:00 | 17.11.2021 07:21 | Night | 470 |
| D-13.12 | 13.12.2021 09:03 | 13.12.2021 10:34 | Day | 50 |
| D-14.12 | 14.12.2021 16:07 | 15.12.2021 08:20 | Night | 230 |
| D-15.12 | 15.12.2021 17:58 | 16.12.2021 08:41 | Night | 130 |

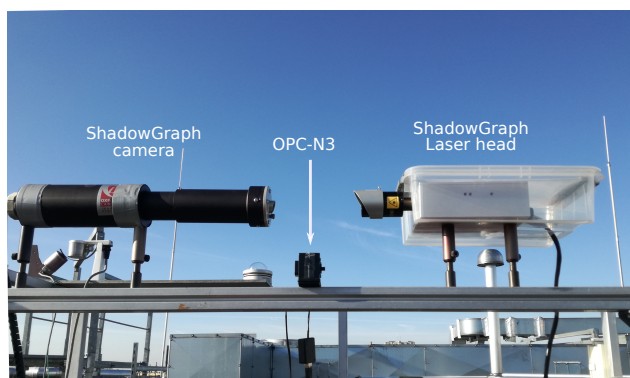

**Figure 1.** OPC-N3 and ShadowGraph at the Radiation Transfer Laboratory, Institute of Geophysics, Faculty of Physics, University of Warsaw, Poland

### 2.1.1 VisiSize D30

The VisiSize D30 is a shadowgraph system manufactured by Oxford Lasers Ltd. The VisiSize D30 (hereafter called Shad-
owGraph) captures the shadow images of particles passing through a measurement volume between a laser head and a high-
resolution camera. It can detect particles of different sizes and shapes in various suspensions in real-time. The images are
acquired by backlight illumination of a measurement volume using an infrared diode laser (808 nm). The camera is triggered
in time with the laser pulses so that a single pulse of laser light occurs in each exposure of the camera. It can capture shadow
images of particles with a maximum rate of 30 frames per second. The size and velocity of the captured particles are then mea-
sured by the ShadowGraph using software provided by the manufacturer based on the Particle/Droplet Image Analysis (PDIA)
method. The PDIA was first introduced by (Kashdan et al., 2003, 2004) to measure industrial sprays. The system allows for
retrieving microphysical properties such as shape, size, $DSD(r)$, total droplet number concentration ($N_c$) and LWC.

The ShadowGraph camera is equipped with different lens magnification settings, and this adjustment can change the reso-
lution of the sample volume. The manufacturer provides calibration for three settings: $\times 1$, $\times 2$, $\times 4$. For this study, the magni-
fication setting of $\times 4$ was used, as it allows for measuring the smallest droplets. The effective pixel size for this magnification

**Table 2.** Technical specifications for the two instruments provided by the manufacturers; by Nowak et al. Nowak et al. (2021) and Hagan et al. Hagan and Kroll (2020).

| **ShadowGraph** | | **Lens setting** | $\times 4$ |
|---|---|---|---|
| Camera sensor [pix × pix] | 1952 × 1112 | Magnification | 6.12 |
| Frame rate [fps] | 30 | Effective pixel size [µm] | 0.90 |
| Laser wavelength [nm] | 808 | Resolution [µm] | 2.0 |
| Laser pulse duration [µs] | 0.1-5.0 | Field of view [mm x mm] | 1.75 x 1.00 |
| One run [min] | 10 | Depth of field [mm] | 5.2 |
| | | Sample volume [cm$^3$] | 0.0092 |
| | | Typical flow rate [cm$^3$ s$^{-1}$] | 0.28 |
| **Alphasense OPC-N3** | | | |
| Particle range [µm] | 0.35-40 | Laser wavelength [nm] | 658 |
| Bins | 24 | Viewing angle | $32° - 88°$ |
| Sampling [s] | 10 | Refractive Index | $1.5 + 0i$ |
| Weight [g] | 105 | Typical flow rate [cm$^3$ s$^{-1}$] | 5.5 |

is 0.9 µm. The ShadowGraph collects data in runs; one run was set to last 10 minutes. There is a pause of around 1 second between runs to save the data. Table 2 presents more details on the ShadowGraph specifications. The ShadowGraph has already been used successfully to study cloud microphysics in both laboratory and in situ measurements. The droplet detection and sizing system of the ShadowGraph were described by Nowak et al. (2021) in detail. The data collected with ShadowGraph on
orographic clouds, i.e. in fog conditions over the mountains, were analyzed by Mohammadi et al. (2022).

### 2.1.2 OPC-N3

OPC-N3 is an optical particle counter developed by Alphasense Ltd. The device is constructed of a diode laser light (wavelength 658 nm) and an elliptical mirror that reflects the laser to a detector. The flow is perpendicular to the laser beam. By using a fan to force the flow, the device can operate continuously without the need to periodically replace the pump filters. OPC-N3
measures the particle number concentration (PNC) in 24 bins, in a diameter range of 0.35 to 40 µm. The PNC measured by the on-board algorithm is converted to PM$_1$, PM$_{2.5}$ and PM$_{10}$. OPC-N3 can measure the mass concentrations of particles up to 2000 µg m$^{-3}$. The sensor returns information about the internal temperature and relative humidity. More information on the Alphasense OPC-N3 specification is provided in Table 2, more details about OPC-N3 can be found in Hagan and Kroll (2020).

## 3 Methods

### 3.1 Fog detection

To automatically retrieve data concerning fog events, calculations were performed to obtain the minimum concentration of droplets needed to create a fog. The Koschmieder formula (Equation 1) relates visibility (VIS) to the extinction coefficient $\sigma$.

$$\text{VIS} = \frac{\ln 50}{\sigma} \tag{1}$$

The extinction coefficient for spherical particles $\sigma$ is defined by the equation:

$$\sigma = \pi \int r^2 Q_e \text{DSD}(r) dr \tag{2}$$

where $r$ - droplet radius, $\text{DSD}(r)$ - droplet size distribution and $Q_e$ is the efficiency coefficient for extinction. To estimate the minimum $N_c$, it was assumed that fog consists of a monodisperse distribution of droplets with a radius of 9.76 μm (effective radius of droplets measured by ShadowGraph). From the Mie theory for the radius of 9.76 μm the $Q_e$ was calculated to be 2.17.

Based on Equations 1) and 2) the minimum concentration of droplets necessary to cause fog was $6.03\,\text{cm}^{-3}$. For calculations of the microphysical properties of fog, data from ShadowGraph for which the droplet concentration is greater than or equal to $6.03\,\text{cm}^{-3}$ were taken into account. During five days, 103 runs (each of 10 min) with detected fog were obtained.

### 3.2 ShadowGraph depth-of-field

When a droplet is at a distance from the camera's focal plane (i.e., a defocused droplet), the generated shadow image consists of a gray halo area around a dark interior circle. If the defocus distance increases, the halo area will grow faster than the total image area. It reaches point where it takes over almost the whole particle image and eventually fades into the background. A criterion was set in the ShadowGraph software algorithm called focus rejection to avoid the sizing error that occurs in the above mentioned situation. The particles with a halo area equal to or larger than $95\%$ of the total particle area are rejected based on this criterion.

To precisely calculate the sample volume in which the droplet can be detected, and subsequently the total droplet number concentration, an accurate measurement of the depth of field (DOF) is essential. The DOF, which is equal to the maximum defocus distance, is assumed by Kashdan et al. (2003) to be linearly dependent on the particle size for a diameter range between 18 and 145 microns.

However, Nowak et al. (2021) showed that for smaller droplets (e.g. cloud-size droplets), the focus rejection criterion plays a key role by imposing a significant constraint on the acceptable DOF. In such cases, the halo area around the droplet exceeds the focus rejection limit at a defocus distance shorter than the linearly size-dependent DOF assumed by Kashdan et al. (2003). Consequently, the relevant sample volume is affected by the focus rejection, which results in an underestimation of the total droplet number concentration.

Nowak et al. (2021) developed a correction method based on the focus rejection criterion in which the DOF for droplet diameter $D_i$ is calculated as follows:

$$z_{95|D_i} = \frac{0.95}{(1 - 0.95a_3)(a_1 D_i + a_2)} \frac{\pi}{4} \frac{(D_i + a_4 + a_5)^2}{pix^2},$$ (3)

where the constants $a_1$ and $a_2$ are calibration constants for a given lens magnification, while constants $a_3$, $a_4$, and $a_5$ refer to the halo blurring, pixelization, and diffraction. In addition, $pix$ represents the effective pixel size equal to 0.9μm for x4 lens magnification setting.

Subsequently, the sample volume for individual droplet $i$ is calculated by multiplying the above DOF and ShadowGraph field of view (FOV) as follows:

$$V_i = 2z_{95|D_i}(L_x - D_i)(L_y - D_i),$$ (4)

where $L_x$ and $L_y$ are respectively the horizontal and vertical length of FOV in μm. Eventually, to calculate the $N_c$, a sum over multiplicative inverse values of $V_i$ must be divided by the total frame numbers $F$ as shown below:

$$N_c = \frac{1}{F} \sum_i \frac{1}{V_i}.$$ (5)

### 3.3 Uncertainty of the measurement

The data from ShadowGraph were integrated every 10 minutes. To calculate the uncertainty of the values (LWC, $N_c$, mean radius) calculated from ShadowGraph, it was assumed that the uncertainty is influenced by the effective pixel size of Shadow-Graph of 0.9 μm and by the Poisson statistics. The formula for the DOF depends on the diameter of the droplet. The uncertainty of a given variable was calculated using the formula for the propagation of uncertainty.

Data from OPC-N3 were collected every 10 seconds, averaged every 10 minutes to compare with ShadowGraph. As the uncertainty of the OPC-N3 data, standard deviation was used, obtained by averaging the data over 10-minute periods and the Poisson statistics.

### 3.4 Scope of compliance between OPC-N3 and ShadowGraph

ShadowGraph samples around 1000 $cm^3$ of air over the period of one hour. To obtain a smooth droplet size distribution spectrum from ShadowGraph, the data needs to be averaged for a period longer than 10 minutes. The minimum averaging time needs to be chosen in such a way that the spectrum is smooth and the time of averaging still allows for the analysis of spectrum changes over time. The data were averaged over one hour[1]. Figure 2 (**a**) shows the volume droplet size distribution (vDSD($r$)) and (**b**) droplet size distribution (DSD($r$)) of OPC-N3 and ShadowGraph. Formulas for DSD($r$) and vDSD($r$) are are given, respectively, as:

$$\text{DSD}(r_b) = N_b \cdot (V_b \cdot \Delta r_b)^{-1},$$ (6)

---

[1]As there is an interval of around 1 second between ShadowGraph runs, the data are averaged over one hour plus around 6-10 seconds

$$\text{vDSD}(r_b) = DSD(r_b) \cdot r_b^3, \tag{7}$$

where $N_b$ number of droplets in a bin, $V_b$ - volume of a bin, $\Delta r_b$ width of the bin, $r_b$ mean bin droplet radius.

Aphasense OPC-N3 manual specifies that OPC-N3 records droplet radiuses in the range of $0.175$ to $20.000$ µm. Fig. 2 shows that the last bin of OPC-N3 registers a much greater mass of droplets than it would appear from the distribution. This indicates that the last bin has not been properly assigned its right edge. In the last bin of OPC-N3 the particles of radiuses larger than 20.00 µm are also counted.

     The last OPC-N3 bin contains a significant part of LWC, therefore it was not excluded from the analysis. However, a new
mean radius for this bin was estimated based on ShadowGraph data. One histogram of DSD was created from all ShadowGraph data of fog events. The mean volume radius ($r_V$) was calculated for droplets bigger than 18.5 µm. The obtained $r_V$ is 21.76 µm. Based on this calculation the lower boundary of OPC-N3 last bin is set as 18.5 µm, the mean radius is set to 21.76 µm and the upper boundary is set as 25.02 µm (the upper limit is set in such a way that the center of the bin falls on the mean radius). Fig. 2 (**c**) shows the vDSD after adjustment of the last bin of OPC-N3.

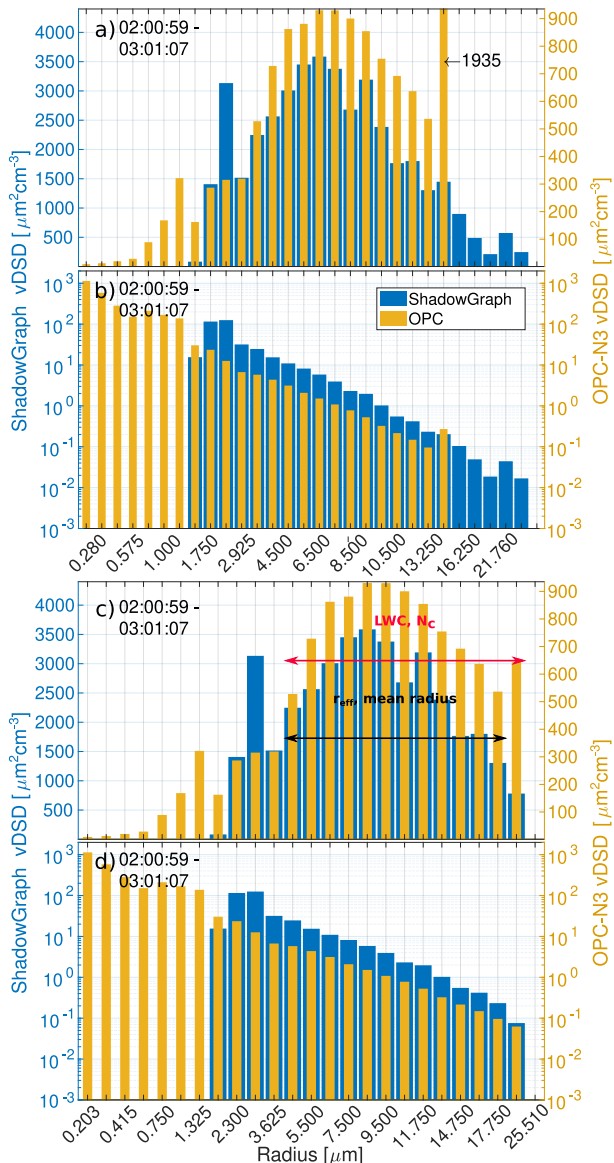

**Figure 2.** The droplet size distribution DSD($r$) - Fig. (**b**) and (**d**) ) and volume droplet size distribution vDSD($r$) - Fig. (**a**) and (**c**) of OPC-N3 and ShadowGraph.The values obtained for each bin of OPC-N3 are shown in yellow, ShadowGraph data, aggregated into the same bins as OPC-N3 are shown in blue. The data were averaged by one hour (November 17, 2021, 02:00:59 - 03:01:07). The right axis corresponds to OPC-N3. Fig. (**a**) and (**b**) represent the bins from OPC-N3 as they are provided by the manufacturer. In Fig. (**c**) and (**d**), the last bin of OPC-N3 has been adjusted and its range is $(18.5 - 25.02]$ µm. In Fig. (**c**), in red the range of OPC-N3 bins taken into account for the calculations of LWC and $N_c$ is shown, and in black for $r_{eff}$ and mean radius. The number 1935 on Fig. (**a**) indicates the value of the last bin of OPC-N3.

Oxford Lasers state that the minimum value of the diameter that can be imaged by the ShadowGraph for a magnification of ×4 is 2 μm. The minimum diameter of fog droplets registered by ShadowGraph was 3.47 μm.

    ShadowGraph spectrum has a peak in the bin of droplets mean radius 2.925μm. This peak is not visible in the OPC-N3 data. It appears to be a problem in recognizing the actual size of small droplets by the algorithm. The pixel size of ShadowGraph is 0.9 μm, leading to a misinterpretation of droplets 2-3 pixels in size. In the article of Kashdan et al. (2003) there is a peak

in the droplet distribution function for the bin with the smallest droplets. This is explained by the misclassification of small unfocused droplets. Identification of the droplets and their size is done by distinguishing the brightness between the shadow of the drop and the brightness of the background pixels. For fuzzy droplets, the difference is small, making it difficult to classify the droplets to the correct size. In our analysis, the droplets below a radius of 4 μm were excluded.

    During the five days of the study, fog was observed for 17.2 hours. Microphysical parameters such as $N_c$, LWC, $r_{eff}$ and

225 mean radius were calculated. For those calculations, droplets from the range of 4 μm to 25.024 μm were taken into account for both devices. In the case of OPC-N3, it corresponds to bins from 12 to 24. Bin 24 was taken into account as it contains a significant amount of LWC, but after adjusting its mean radius to 21.76 μm. Droplets smaller than 4 μm were excluded from the comparison as ShadowGraph has problems correctly classifying them.

    For the comparison of $r_{eff}$ and mean radius, only the droplets from the range of 4 μm to 18.5 μm were taken into account

for ShadowGraph and OPC-N3 (bins 12 to 23 of OPC-N3), as in this range, the registered sizes of droplets from both devices overlap without any adjustments. The droplet radius range, taken into account for each microphysic parameter, is shown in Appendix A Table A1.

    Fig. 2 shows that the values of DSD for OPC-N3 are smaller than for ShadowGraph. Appendix A presents a study regarding whether correcting the OPC-N3 data based on RI allows for better data compliance with ShadowGraph.

Particles in OPC-N3 are assigned to the size bins based on the amount of light scattered. In Mie theory, the amount of light scattered can be linked to a specific size of a particle; to be able to use Mie theory, one needs to assume an appropriate refractive index (RI) of a particle. In OPC-N3, the manufacturer set the $\mathrm{RI}_{OPC}$ at $1.500 + 0.00i$. The refractive index of pure water - $\mathrm{RI}_{water} = 1.331 + 0.000i$ (Hale and Querry, 1973) - is different from the one assumed in OPC-N3, which can lead to an incorrect assignment of the droplets to the correct bin based on their radius.

We analyzed the effect of recalculating the assignment of droplets to specific bins using $\mathrm{RI}_{water}$. As a result, we found out that the vDSD spectrum of OPC-N3 and ShadowGraph after RI correction does not overlap as well as when it was assumed $\mathrm{RI}_{OPC}$, however, due to the greater representation of bigger droplets, a better comparison of LWC between the devices was obtained. In this study, the refractive index is the default OPC-N3 $\mathrm{RI}_{OPC}$ in the Appendix the comparison to values calculated using $\mathrm{RI}_{water}$ is shown.

## 3.5   OPC-N3 temperature and relative humidity

Alphasense OPC-N3 reports temperature ($T$) and relative humidity (RH). The sensors are mounted inside the device. The registered values were compared with Vaisala WXT520, which is mounted on the roof of the Institute of Geophysics and measures ambient $T$ and RH. As seen in Fig. 3 (**a**) the $T$ inside OPC-N3 is approximately $6°$C higher than that of Vaisala

WXT520. The circuit board system is most probably heating the inside of the device. This causes a drop in humidity within OPC-N3. When comparing the RH of OPC-N3 with Vaisala WXT520, it is seen that the RH within OPC-N3 is lower by around 20 %. The measurements presented were taken during the night and early morning. During the night, the value from OPC-N3 seems to be shifted by a constant value in comparison with Vaisala WXT520. The temperature and RH registered by OPC-N3 rapidly change after sunrise (6:00 UTC). The OPC-N3 is made of black plastic, which is probably heated by sunlight; the higher temperature of the body affects the inside humidity. This leads to the conclusion that the $T$ and RH reported by OPC-N3 cannot be used as the values of ambient conditions and cannot be easily corrected. Furthermore, a higher $T$ and lower RH within OPC-N3 may impact on the droplet size due to the evaporation process.

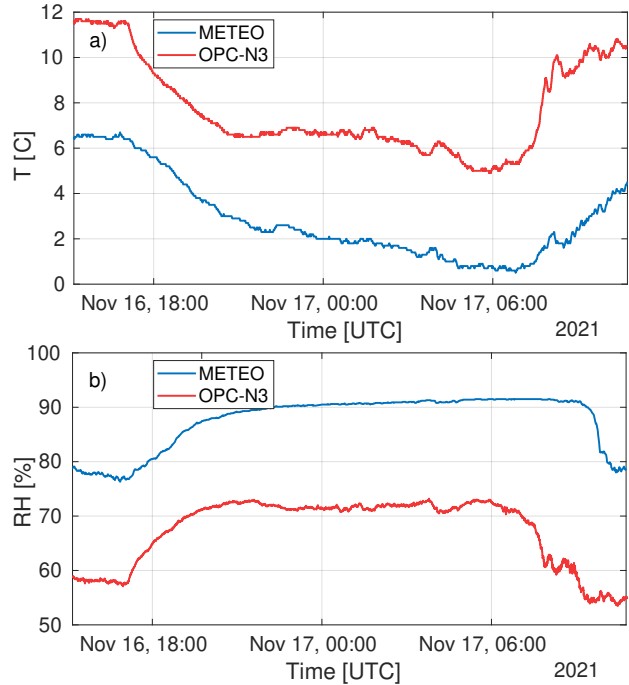

**Figure 3.** Comparison between OPC-N3 (red line) with Vaisala WXT520 (blue line) of (**a**) temperature ($T$) and (**b**) relative humidity (RH). Devices mounted on the roof of the Institute of Geophysics, sunrise at 6:00 UTC.

## 4 Results

### 4.1 Microphysical properties of fog

The data obtained from all the dates of fog occurrence (listed in Table 1) were used for comparison of microphysical properties of fog between OPC-N3 and ShadowGraph.

### 4.1.1 Liquid water content

LWC results are shown in Fig. 4 (**a**). The LWC calculated from ShadowGraph ranges from 0 to $0.2 \ \mathrm{g \, m^{-3}}$. The comparison between OPC-N3 and ShadowGraph shows a linear relationship between devices. The OPC-N3 values are almost two times lower than those from ShadowGraph. The Pearson correlation coefficient (PCC) is $0.92$.

Values such as LWC, $N_c$, droplet diameter statistics obtained from OPC-N3 can be corrected using a linear regression curve of the form $y = ax + b$, where $x$ is the measured value from OPC-N3, $a$ and $b$ are calculated coefficients and $y$ is the measured value from ShadowGraph. The linear regression coefficients, PCC and RMSE for LWC are presented in Appendix B Table B1.

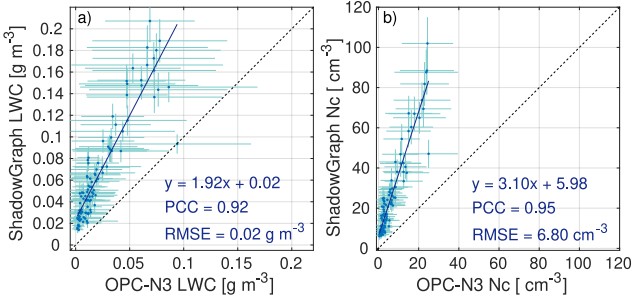

**Figure 4.** Comparison between ShadowGraph and OPC-N3 of (**a**) liquid water content (LWC) (**b**) total droplet number concentration ($N_c$). Each point represents a value measured over a 10 minute period. Blue points represent data retrieved from the devices, solid deep blue lines represent linear fit to the data, and bars represent the estimation of uncertainty.

### 4.1.2 Total number concentration

The droplet number concentration is shown in Fig. 4 (**b**), the values span from below 10 to around $100 \ \mathrm{cm^{-3}}$ for ShadowGraph.
There is a high Pearson correlation coefficient equal to 0.95 for the comparison with OPC-N3. The values obtained from OPC-N3 are lower than those obtained from ShadowGraph. The linear regression coefficients, PCC and RMSE for $N_c$ are presented in Appendix B Table B1.

### 4.1.3 Droplet diameter

Four different droplet diameter statistical moments were calculated. The arithmetic mean $r$, the surface mean $r_S$, the volume
mean $r_V$ and the effective radius $r_{eff}$. The values were calculated using a formula for statistical moments. To calculate $r_{eff}$, the following formula was used:

$$r_{eff} = \left( \sum_{i=1}^{m} r_i^3 \cdot N_i \right) \cdot \left( \sum_{i=1}^{m} r_i^2 \cdot N_i \right)^{-1}, \tag{8}$$

where $N_i$ in the case of OPC-N3 is the concentration of droplets in the volume of the bin, $m$ denotes the number of bins. For ShadowGraph, $m$ denotes the number of droplets and $N_i = (FV_i)^{-1}$ where $V_i$ is the volume in which the droplets were

280 measured. To calculate the mean radius $r$ the following formula was used:

$$r = \left( \sum_{i=1}^{m} r_i \cdot N_i \right) \cdot \left( \sum_{i=1}^{m} N_i \right)^{-1}, \tag{9}$$

To calculate the mean surface radius $r_S$ the following formula was used:

$$r_S = \left( \left( \sum_{i=1}^{m} r_i^2 \cdot N_i \right) \cdot \left( \sum_{i=1}^{m} N_i \right)^{-1} \right)^{1/2}, \tag{10}$$

To calculate the mean volume radius $r_V$ the following formula was used:

$$r_V = \left( \left( \sum_{i=1}^{m} r_i^3 \cdot N_i \right) \cdot \left( \sum_{i=1}^{m} N_i \right)^{-1} \right)^{1/3}, \tag{11}$$

Figure 5 shows the $r_{eff}$ and Fig. 6 (**a**) the mean radius, (**b**) the mean surface radius and (**c**) the mean volume radius. The $r_{eff}$ is between 8.05 and 11.71 μm for ShadowGraph. There is a modest correlation coefficient between the devices for all mean radiuses. The linear regression coefficients, PCC and RMSE are presented in Appendix B Table B1.

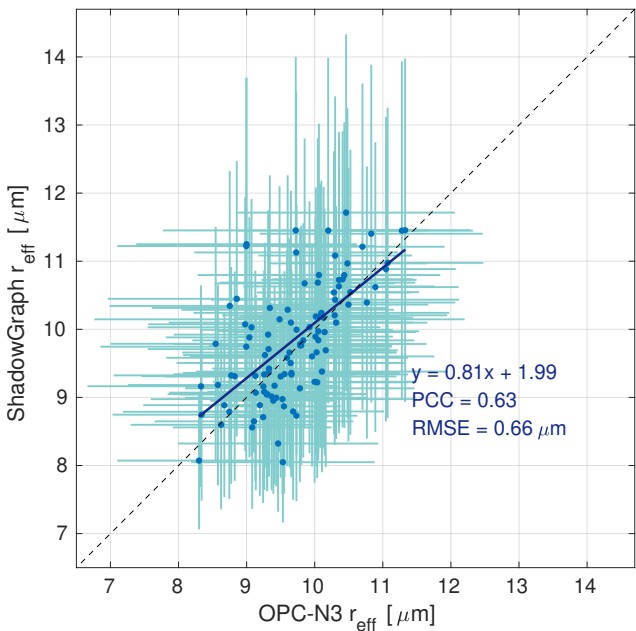

**Figure 5.** Comparison between ShadowGraph and OPC-N3 of $r_{eff}$ [μm]. Each point represents a value averaged over 10 minutes, and the bars represent the estimation of uncertainty. Blue points and lines represent data retrieved from the devices, and solid deep blue lines represent the linear fit to the data.

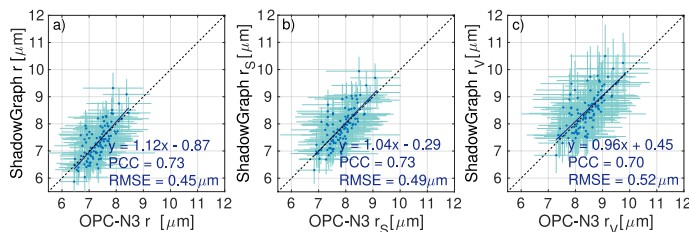

**Figure 6.** Comparison between ShadowGraph and OPC-N3 for (**a**) mean radius [µm], (**b**) mean surface radius [µm], (**c**) mean volume radius [µm]. Each point represents a value averaged over 10 minutes, and the bars represent the estimation of uncertainty. Blue points and lines represent data retrieved from the devices, and solid deep blue lines represent linear fit to the data.

## 5    Case study

During the measurements, five fog events were registered. Most of them lasted less than two and a half hours. The longest event occurred at night from November 16 to 17, 2021. As this case of fog lasted for almost eight hours, it was chosen for the case study in this section.

### 5.1    Overview of weather conditions

On November 16, 2021, Poland was in the saddle region, between two highs over the North Atlantic and Ukraine. The anticyclone above Ukraine moved towards the Caspian Sea, causing the pressure to drop from 1008 hPa at 18:00 UTC to 1004 hPa around 07:00 UTC on November 17, 2021. Atmospheric conditions at the saddle point resulted in a decrease in wind speed during the measurements. The wind blowing from the direction of $150°$ was on average $2.2\,\mathrm{m\,s^{-1}}$, which favoured the formation of fog.

Fog started to form around 21:15 UTC. The sky was clear before fog occurrence (infrared incoming surface flux was around $250\,\mathrm{W\,m^2}$ before 20 UTC). Infrared flux started to rise around 21 UTC, and dropped around 22:16, suggesting a temporary disappearance of fog. At approximately 4:00 UTC, the infrared flux started to decrease, and after sunrise the total flux increased - the line representing the total flux is jagged, which suggests that in the said morning, there was a layer of thin clouds above the fog.

From 18:00 UTC, on November 16, 2021, the air temperature decreased from $5.4°C$ to a minimum of $0.5°C$ just before 07:00 UTC the next day. The temperature then started to increase, and the process of dissipating the fog started. Appendix C Fig. C1 shows the atmospheric situation on the measurement site in detail and Appendix C Fig. C2 presents the radiosounding from 00 UTC November 17, 2021 from Legionowo, which is near Warsaw city.

In Appendix C Fig. C1 (**a**) shows the scattering coefficient at 525 nm measured by Aurora 4000 nephelometer at dry conditions (the humidity inside Aurora 4000 during the measurements was below 30 %). The values increased from $178\,\mathrm{Mm^{-1}}$ at 18:00 UTC to 250 at 19:30 UTC, then oscillated between 250-260 $\mathrm{Mm^{-1}}$ until 23:15 UTC, then continuously decreased to 170 $\mathrm{Mm^{-1}}$ at 07:00 UTC. Values above 200 $\mathrm{Mm^{-1}}$ suggest moderate smog conditions. The increase in the scattering

coefficient after 18:00 UTC was probably due to air pollution caused by the activation of the house heating system and traffic emissions.

The PM values obtained from OPC-N3 are presented in Fig. 7. The $PM_1$ was around $17\,\mu g\,m^{-3}$ at 18:00 UTC and increased to $125\,\mu g\,m^{-3}$ at 23:00 UTC, then the $PM_1$ value decreased. $PM_{10}$ data exhibit a different behaviour from PM1. Between 21:00 and 22:30 UTC, 22:45 and 04:00 UTC, 04:20 and 06:10 UTC, episodes of significant increase and decay in $PM_{10}$ value were observed. For each episode, $PM_{10}$ reached, respectively $(14960, 12550, 6897\,\mu g\,m^{-3})$. A similar pattern was observed in $PM_{2.5}$ but less pronounced $(1354, 1148, 893\,\mu g\,m^{-3})$. Such values are unrealistic for the City of Warsaw, the values are biased due to high humidity and fog droplet contamination. These episodes correspond to three periods of fog events. In Fig. 7 (**b**) the estimated values of LWC are presented for OPC-N3 and ShadowGraph.

The data collected from OPC-N3 allows to calculate the effective radius. Fig. 7 (**b**) shows the effective radius calculated from the droplets that have a radius bigger than $1.15\,\mu m$ ($r_{eff}^{1.15}$). The $r_{eff}^{1.15}$ rapidly increases after the fog onset. When there are high concentrations of $PM$, based on $r_{eff}^{1.15}$, it is possible to distinguish an episode of haze from an episode of fog.

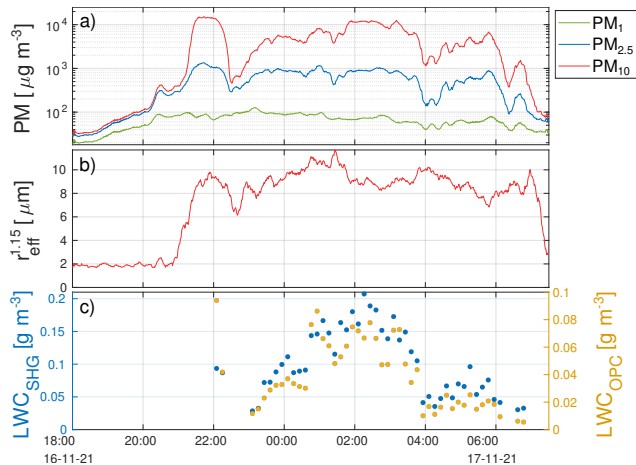

**Figure 7.** Situation during the night of November 16, 2021: (**a**) $PM_1$, $PM_{2.5}$ and $PM_{10}$ obtained from Alphasesnse OPC-N3. The data from this instrument was collected every 10 seconds, and later a running mean over a ten minute window was applied to smooth the data. (**b**) Effective radius calculated from OPC-N3 for droplets (radiuses from $1.15\,\mu m$ were taken into account). (**c**) LWC obtained from ShadowGraph and OPC-N3. The values were calculated every ten minutes. LWC was calculated only for fog occurrence (automatic fog detection was described in Section 3.1).

## 5.2   Evolution of the droplet spectrum

In this subsection, the evolution of the droplet spectrum is shown for the period between 22:00 UTC on November 16 and 07:00 UTC on November 17. Figure 8 shows the vDSD$(r)$ for OPC-N3 and ShadowGraph. In one plot, the data for the average of one hour are shown.

Figure 8 shows a relatively good agreement between devices on the behaviour of the droplet spectrum. During the night of November 16, the first fog event occurred from 21:00 to 22:20 UTC. Due to some connection errors, the ShadowGraph was set at 22:00 UTC. Moreover, during the first 10 minute run, it did not recognise all the droplets that passed through it, probably due to the steamed lens.

The spectrum obtained from OPC-N3 consists of a two-mode distribution. The first mode has a maximum for bin mean radius 1.33 μm and its location remains steady at night. The second mode peaked during hours 23:00 - 02:00 UTC moves from 11.75 μm to 14.75 μm. During this time, the fog evolved and the value of the vDSD($r$) increased. The maximum intensity of the fog was observed at 02:00 - 03:00 UTC and the peak of the second mode rapidly changed to 8.50 μm. After 3:00 UTC, the second mode of droplet distribution flattened out and the fog started to disappear. The data obtained from ShadowGraph followed the same pattern and the values obtained from ShadowGraph were greater. From 02:00 to 03:00 UTC, the maximum value of the vDSD($r$) was 930 $\mu m^2\,cm^{-3}$ for OPC-N3 and 3586 $\mu m^2\,cm^{-3}$ for ShadowGraph.

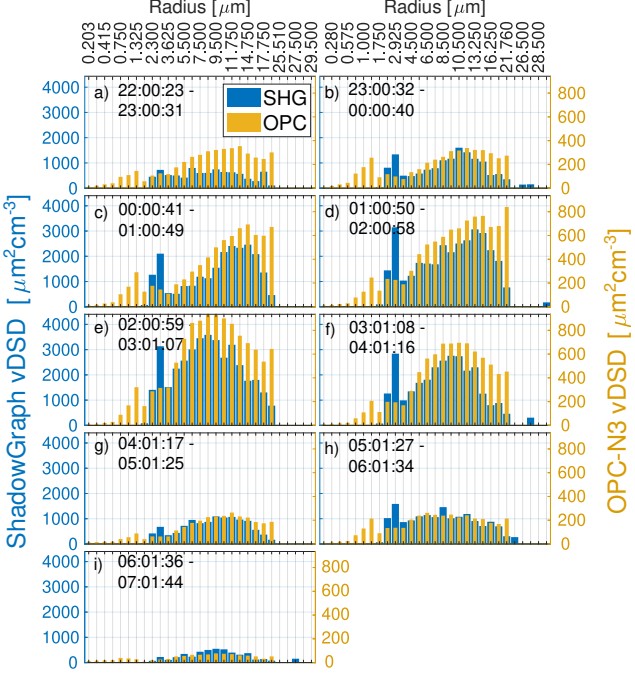

**Figure 8.** Volume droplet size distribution (vDSD($r$)). Data obtained from ShadowGraph are shown in blue, from OPC-N3 in yellow. Figures from (**a**) to (**i**) represent consecutive hours between 22:00 UTC on November 16 and 07:00 UTC on November 17.

## 6    Conclusions

The low-cost optical particle counter Alphasense OPC-N3 was tested for its application in the measurement of fog droplets and compared to Oxford Lasers VisiSize D30 ShadowGraph. The following results are obtained from the current work.

- In this article, we have shown that OPC-N3 registers fog droplets. Badura et al. (2018) showed that the previous version of Alphasense OPC overestimates PM values in high humidity. Our results are in line with this article, OPC-N3 in the case of high humidity $RH = 80-90\,\%$ can overestimate PM values. During this study, the value of $PM_{10}$ reached above $15\,\mathrm{mg\,m^{-3}}$. Such high values cannot be explained only by the higroscopic growth of aerosols. The high PM values are associated with fog occurrence and are a result of the count of fog droplets by OPC-N3. The main point of this article was to verify the hypothesis that the values are overestimated because OPC-N3 counts fog droplets.

- OPC-N3 can be used to automatically detect the fog. After fog onset, the effective radius for particles bigger than aerosol (only droplets from bins 7 and higher taken into account) drastically increases (from a value of 2 μm to around 6-10 μm) and stays high for the duration of the fog.

- The manual of OPC-N3 provides information that OPC-N3 counts droplets diameters up to 20 μm. However in this study we observed that the last bin (diameter $18.5-20$ μm) of OPC-N3 counts not only drops to 20 μm, but also larger. Because this instrument is used mainly for aerosol measurements, which are usually not present at this size, this problem has not been described before. This fact was possible to observe by presenting the results from ShadowGraph and OPC-N3 as the volume droplet size distribution vDSD($r$). We estimated based on ShadowGraph data that the last bin of OPC-N3 for this analysis collects droplets from $18.5-25.02$ μm.

- The minimal practical radius that can be measured by ShadowGraph is approximately 3.5-4 μm for lens magnification $\times 4$. The manual provided by Oxford Lasers reports that the minimum radius resolution of ShadowGraph is 1.01 μm. However, during the measurements, ShadowGraph did not record droplets smaller than 1.73 μm. Interestingly, the results reveal that there were non-physical concentrations of droplets in bin (2.6 -3.24 μm). As we stated in Sec. 3.4 this peak probably comes from noise and problems with the wrong diameter assignment of small defocused droplets. As there were some limitations in comparing the radius ranges of both devices, the range of radius taken for comparison in this article was from 4 μm to 18.5 / 25.02 μm.

- The LWC, $N_c$, $r_{eff}$ and the statistical moments of radius were calculated based on the data obtained from OPC-N3. The values were compared with those retrieved from ShadowGraph. Overall, these results suggest that there is a high correlation between OPC-N3 and ShadowGraph. OPC-N3 can be used to measure fog microphysics; however, the values reported by OPC-N3 need to be corrected.

- We have shown that the measured $T$ and $RH$ by OPC-N3 are biased. As shown in Section 3.5 OPC-N3 $T$ is higher by $6^{\circ}C$ and $RH$ is lower by 20 % compared to ambient $T$ and $RH$. After sunrise, those values change, probably because of device case heating by solar radiation, and because of this it is not possible to propose a correction of the measured $T$ and $RH$ by OPC-N3.

- The OPC-N3 lowers the values compared to ShadowGraph; in the case of LWC the values are 1.92 times lower and in the case of $N_c$ 3.10 times lower. As OPC-N3 uses a default RI index for radius calculations, it was investigated whether

applying the correction of the RI index would improve the compatibility of the instruments (this issue was addressed
in Appendix A). After correction, an improvement in LWC was observed; OPC-N3 data were 0.88 times smaller than
the ShadowGraph values. However, applying the RI correction caused the shifting of the assigned radius of droplets
registered by OPC-N3 to higher values. This resulted in a better compliance between OPC-N3 and Shadowgraph in
LWC, however it decreased compliance in the effective radius. Applying RI correction does not appear to improve the
consistency of the results. It might be that Alphasense has a built-in processing of the data while assigning it to the
bin size. The underestimated results from OPC-N3 appear to be due to the fact that OPC-N3 counts fewer droplets
than the ShadowGraph and not from their wrong assignment to the correct bin size. The reason for less droplets maybe
their evaporation due to the fact that inside the OPC-N3 is higher temperature and lower humidity than environmental
values. The second possibility could be that the open path inside of OPC-N3 allows the flow to expand and this changes
the droplet number concentration. The third reason could be that when the fan speed, which drives the flow inside the
OPC-N3 changes, the algorithm does not properly adjust the amount of sampled air.

OPC-N3, due to its dimensions and low weight, can be mounted on UAVs, cable cars or balloons. Such works have been
performed with small OPCs by Posyniak et al. (2021); Habeck et al. (2022); Girdwood et al. (2020). The main objective of
that research was to investigate changes in PM in the vertical profile. Our work presents a new applicability of OPC-N3 within
UAVS (Unmanned Aerial Vehicles) to measure the vertical profiles of the microphysical properties of fog. Another application
of our study could be to produce inexpensive fog detection systems in airports or automatic fog monitoring systems on roads.
Taking the effective radius of droplets received from OPC-N3 into consideration, it is possible to differentiate the low-visibility
situations between fog conditions (which are not hazardous for people) from haze events, when highly polluted air can cause
health risks to people.

*Code and data availability.* The results presented in this study were obtained with the use of the Oxford Lasers VisiSize D30 software
version 6.5.39 and code developed by the authors in the MATLAB environment. The latter are available from the authors upon request. The
data presented in this study are available from the authors on request.

## Appendix A

### A0.1 Correction of OPC-N3 due to refractive index

Optical particulate matter devices operate by measuring the light scattered by a particle. The Mie theory allows one to relate
the amount of light scattered by a particle to its size, assuming particle sphericity. OPC-N3 has an elliptical mirror that sums
up the amount of light scattered between angles $32 - 88°$ (Hagan and Kroll, 2020).

The proposed correction of the refractive index changes the bin edges of OPC-N3. First, the theoretical scattering cross
section ($C_{scat}$) was calculated for two RIs, one assumed in the OPC-N3 and one for water. The theoretical values of Mie
scattering were calculated for 6000 points of logarithmic spaced radiuses. The curve was then smoothed by running the mean
function (the mean was taken over 100 points). Later, for each mean radius of an OPC-N3 bin ($r_{old}$), $C_{scat}$ was found on the
calibration curve derived for $\mathrm{RI}_{OPC}$ (brown line with red dots in Fig. A1). Then the obtained $C_{scat}$ values were transfered on
the curve for $\mathrm{RI}_{water}$ (deep blue line) and their corresponding radius ($r_{new}$) - blue points in Fig. A1. In the following figures,
the microphysical properties were calculated for the uncorrected radiuses $r_{old}$ and the corrected ones $r_{new}$.

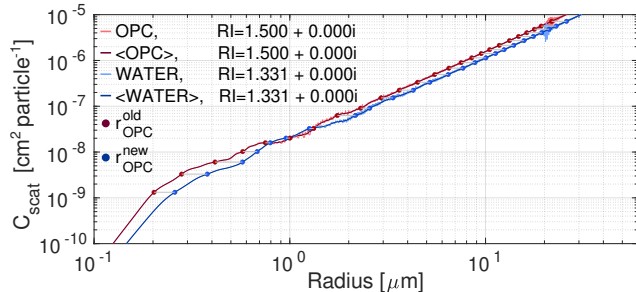

**Figure A1.** Theoretical scattering cross section $C_{scat}$ calculated by the Mie theory for refractive index assumed by Alphasense in OPC-N3
($1.500 + 0.00i$) and RI for pure water ($1.331 + 0.000i$) integrated for scattering angles $32° - 88°$. Curves were smoothed by running mean
(lines denoted as <OPC> and <WATER>). Points indicate the $C_{scat}$ values at each middle radius of the bin (red dots) and blue dots for
corrected radius values assuming RI of water.

In the main text, droplets with radiuses from 4 µm to 18.5 µm (for $r_{eff}$, $r$, $r_s$, $r_V$) or 25.02 µm (for LWC, $N_c$) were taken
into account. In this section, we show figures that compare the data not corrected ($\mathrm{RI}_{OPC}$, represented in blue) to the data after
the RI correction ($\mathrm{RI}_{water}$, depicted in red). In the case of $\mathrm{RI}_{water}$, the bin edges of OPC-N3 where shifted. The edge 18.5 µm
was shifted to 22.03 µm and 25.02 µm was shifted to 28.31 µm. In the case of $\mathrm{RI}_{water}$, the range of droplet radius taken into
account for comparison between OPC-N3 and ShadowGraph was from 3.82 µm to 23.51 µm (for $r_{eff}$, $r$, $r_s$, $r_V$ or 28.31 µm
(for LWC, $N_c$), the appropriate ranges are listed in Table A1.

**Table A1.** The range of droplet radius which was taken into account for the calculation of microphysical properties of fog.

| droplet radii | LWC, $N_c$ | $r_{eff}, r, r_s, r_V$ |
|---|---|---|
| no RI corr. (RI$_{OPC}$): | 4.00 - 25.02 [µm] | 4.00 - 18.50 [µm] |
| RI corr. (RI$_{water}$): | 3.82 - 28.31 [µm] | 3.82 - 23.51 [µm] |

## A0.2   Microphysical properties of fog

LWC results are shown in Fig. A2 (**a**). Applying the RI correction reduces the difference in LWC between devices; OPC-N3 shows values 0.88 smaller than those from ShadowGraph. The droplet number concentration is shown in Fig. A2 (**b**), the proposed RI correction does not make a significant improvement in the $N_c$ data. In the case of LWC and $N_c$ PCC and RMSE are similar regardless of whether the correction was applied or not. The linear regression coefficients, PCC and RMSE for LWC are presented in Appendix B Table B1.

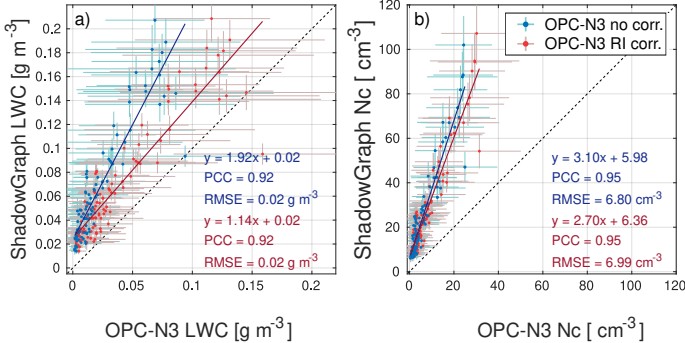

**Figure A2.** Comparison between ShadowGraph and OPC-N3 of (**a**) liquid water content (LWC) (**b**) total droplet number concentration ($N_c$). Each point represents a value measured over 10 minutes. Solid blue and red lines represent linear fit to the data. Blue points and lines represent data retrieved from the devices, red points and lines represent data after correction of the refractive index for OPC-N3.

The $r_{eff}$ is shown in Fig. A3. The correction of RI index is applied only to OPC-N3, however, the RI correction affects the bin boundaries, which also has an impact on the data from ShadowGraph (the droplet radius range taken into account is wider after RI correction)

Applying the RI correction caused the shifting of the assigned radiuses of droplets registered by OPC-N3 compared to ShadowGraph (Fig. A3). The RMSE is higher between OPC and ShadowGraph than without correction. As the droplets are assumed to be larger, the LWC obtained is higher.

Figure A4 shows the evolution of the vDSD during D-16.11 fog case. The vDSD obtained from OPC-N3 is less consistent with the vDSD obtained from ShadowGraph than it was in the case without RI correction. The OPC-N3 vDSD consists of two modes, one with a peak around 0.8 µm and other around 14 µm. The second mode is shifted towards bigger droplet diameters compared to the distribution from ShadowGraph. Applying such a correction does not appear to improve the consistency of

the results. It might be that Aphasense has a built-in processing of the data while assigning the bin size. The underestimated results from OPC-N3 appear to be due to the fact that OPC-N3 counts fewer droplets than the ShadowGraph and not from their wrong assignment to the correct bin size.

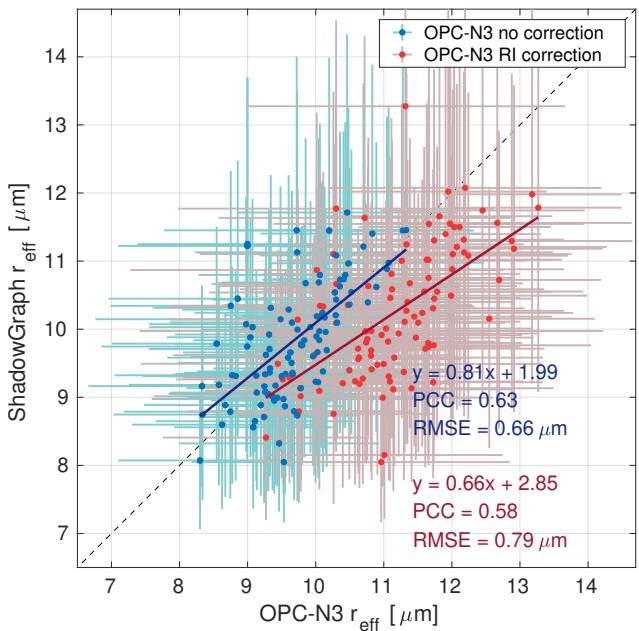

**Figure A3.** Comparison between ShadowGraph and OPC-N3 of $< r_{eff} >$ [ μm]. Each point represents values averaged over ten minutes, the bars represent the estimation of uncertainty. Solid blue and red lines represent linear fit to the data. Blue points and lines represent data retrieved from the devices, red points and lines represent data after correction of the refractive index for OPC-N3.

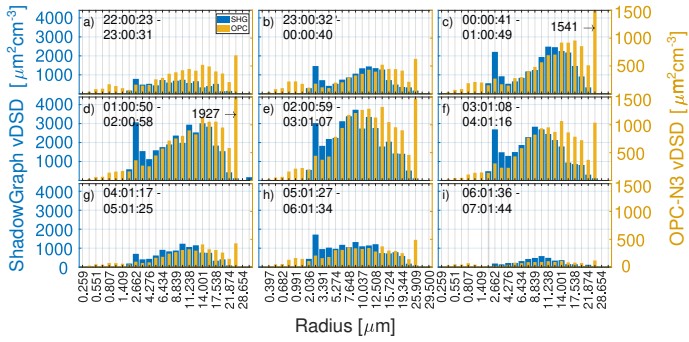

**Figure A4.** Volume droplet size distribution (vDSD($r$)) after applying RI correction to the data from OPC-N3. The bin mean radiuses are different in comparison to Fig. 8. The data obtained from ShadowGraph are shown in blue, from OPC-N3 in yellow. Figures from (**a**) to (**i**) represent consecutive hours between 22:00 UTC on November 16 and 07:00 UTC on November 17. The bin value which is out of range has an assigned numerical value.

## Appendix B

**Table B1.** The linear regression coefficients, Pearson's correlation coefficient (PCC) and the root mean square error (RMSE) for LWC, $N_c$, $r_{eff}$, $r$, $r_s$, $r_V$. The linear regression was obtained for data without RI correction and with RI correction applied. The linear regression curve of the form y = ax + b was fitted to the respective quantity, where x is the measured value from OPC-N3, a and b are calculated coefficients and y is the measured value from ShadowGraph.

| | | No RI correction | | RI correction | | |
|---|---|---|---|---|---|---|
| LWC | $a$ | $1.921 \pm 0.084$ | | $a$ | $1.144 \pm 0.050$ | |
| | $b$ | $0.024 \pm 0.003$ | $[\mathrm{gm}^{-3}]$ | $b$ | $0.025 \pm 0.003$ | $[\mathrm{gm}^{-3}]$ |
| | PCC | 0.916 | | PCC | 0.916 | |
| | RMSE | 0.019 | $[\mathrm{gm}^{-3}]$ | RMSE | 0.020 | $[\mathrm{gm}^{-3}]$ |
| $N_c$ | $a$ | $3.100 \pm 0.106$ | | $a$ | $2.702 \pm 0.089$ | |
| | $b$ | $5.979 \pm 0.930$ | $[\mathrm{cm}^{-3}]$ | $b$ | $6.361 \pm 0.955$ | $[\mathrm{cm}^{-3}]$ |
| | PCC | 0.945 | | PCC | 0.949 | |
| | RMSE | 6.803 | $[\mathrm{cm}^{-3}]$ | RMSE | 6.991 | $[\mathrm{cm}^{-3}]$ |
| $r_{eff}$ | $a$ | $0.810 \pm 0.099$ | | $a$ | $0.663 \pm 0.092$ | |
| | $b$ | $1.994 \pm 0.960$ | [μm] | $b$ | $2.848 \pm 1.032$ | [μm] |
| | PCC | 0.632 | | PCC | 0.583 | |
| | RMSE | 0.657 | [μm] | RMSE | 0.791 | [μm] |
| $r$ | $a$ | $1.118 \pm 0.104$ | | $a$ | $0.894 \pm 0.084$ | |
| | $b$ | $-0.874 \pm 0.765$ | [μm] | $b$ | $0.137 \pm 0.670$ | [μm] |
| | PCC | 0.730 | | PCC | 0.726 | |
| | RMSE | 0.453 | [μm] | RMSE | 0.478 | [μm] |
| $r_s$ | $a$ | $1.041 \pm 0.098$ | | $a$ | $0.837 \pm 0.082$ | |
| | $b$ | $-0.294 \pm 0.772$ | [μm] | $b$ | $0.643 \pm 0.708$ | [μm] |
| | PCC | 0.727 | | PCC | 0.714 | |
| | RMSE | 0.488 | [μm] | RMSE | 0.531 | [μm] |
| $r_V$ | $a$ | $0.955 \pm 0.096$ | | $a$ | $0.77 \pm 0.083$ | |
| | $b$ | $0.446 \pm 0.808$ | [μm] | $b$ | $1.358 \pm 0.780$ | [μm] |
| | PCC | 0.705 | | PCC | 0.680 | |
| | RMSE | 0.523 | [μm] | RMSE | 0.587 | [μm] |

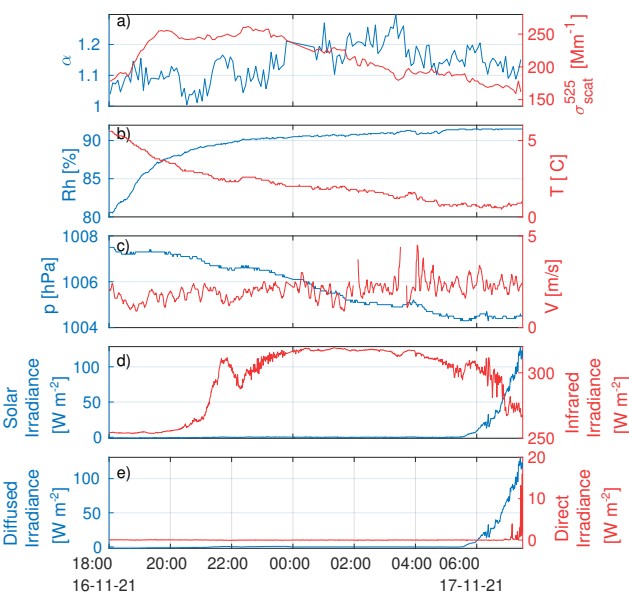

**Figure C1.** Situation during the night of November 16, 2021: (**a**) Scattering coefficient ($\sigma_{scat}^{525}$ and Ångström exponent obtained from Ecoteh Aurora 4000. The data from this instrument was collected every five minutes. (**b**) Temperature ($T$) and relative humidity (RH) obtained from Vaisala WXT520. The data from this instrument was collected every minute. (**c**) Local pressure ($p$) and mean wind velocity ($V$) obtained from Vaisala WXT520. (**d**) Total and infrared irradiance, (**e**) diffused and direct irradiance. The data came from apparatus mounted in the Radiation Transfer Laboratory, Faculty of Physics, University of Warsaw, Poland.

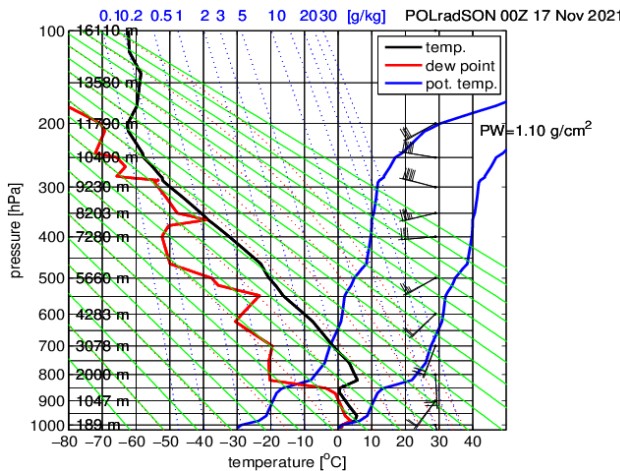

**Figure C2.** Radio sounding from 00 UTC November 17, 2021 from Legionowo (near Warsaw), Poland.

*Author contributions.* SM and KM planned the campaign; KN, MM, KM performed the measurements; KN analyzed the data; KN wrote the manuscript draft; KN, MM, SM, KM reviewed and edited the manuscript.

*Competing interests.* Some authors are members of the editorial board of journal Atmospheric Measurement Techniques. The peer-review process was guided by an independent editor, and the authors have also no other competing interests to declare.

*Acknowledgements.* This research was funded by the National Science Centre (NCN), Poland, grant number UMO-2017/27/B/ST10/00549.

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
