# Peer review of "Applicability of the Low-Cost Optical Particle Counter OPC-N3 for Microphysical Measurements of Fog"

_Atmospheric Measurement Techniques, 2022_

## Referee Comment (RC2)

Applicability of the low-cost optical particle counter OPC-N3 for microphysical measurements of fog.

Katarzyna Nurowska

Dec 02 2022

Decision: Major revisions

General

This paper studies OLV and OPCN3 measurements for fog studies. Compare the measurements after some adjustments for RefIndex and bin intervals. They claimed that OPC can be used for drone applications although significant diff exists, but they never used a drone for this application. Overall significant diff exists between OPC and OLV obs. OLV measures higher spectral densities for a given bin and based on shadow detecting technique versus OPC optical counts. OPC values are corrected based on OLV fit data. OLV measured higher Nd values compared to OPC that leads to higher LWC. Overall manuscript focuses on a good subject but many issues exist based on text flow, analysis, results and conc etc. I feel manuscript needs to be more focused. I suggest major corrections and they are given below.

Major issues

1. Abstract

Needs grammatical corrections such as proprieties…..

Within drones…..using drones, I don't like the drone name change it to UAS or UAV.

Last sentence is not proven and not used. Take it out.

OPC should be corrected based on "prof equipment" doesn't make sense, and how do you know OLV is the best?

Pearson corr coeff? This is for what? Not clear in the text. Why you used <xx>, take it out and use overbar for mean but not needed here.

Issues exist with sentence structures.

INTRODUCTION

After 1 sentence add Gultepe et al 2008 (AMS and Gultepe et al Atmos Res 2017) for warm fog and ice fog definitions.

There are diff types of fog……. Change to this.

L26The cold surface cools……. This sentence not right, needs corrected.

Li23; it to Fog

Line 23-25, very rough sentence, fog affects not only aviation but others…. Check the work.

Ln28; SO2 changes related to fog?

Ln30; liquid??? PM is solid particles not liquid…

Ln34; urban aerosols to marine ones??? Are you sure?

Ln35; increased LWC with smaller droplers? Usually LWC is fixed.

Ln37; last sentence, decrease in LWC??? Not right, remove this sentence.

Intro is not designed properly, for ex, climate info given in diff locations.

Ln49; it is for aerosols but not for fog, correct this….

Ln58; remove this sentence, not written properly.

Ln60; not really correct, usually LWC and Nd or LWC and MVD (see Gultepe et al 2021, Marine fog book)

Ln60 references please

23 channels; not correct, it is adjustable (see Gultepe et al 2008 AMS bull)

Ln65 and 66; allows for consistent comparisons based on various field observations.

Ln68; see Gultepe et al microphysics review in BLM that uses a gondola with CDP2 and BCP, should be ref here.

Ln80; clear sky , no fog, remove this sentence.

Ln85; see Gultepe et al for Radiometerics MWR for fog research (AMS Bull 2008; Atmos Res 2017, Gultepe at al)

Ln92-93; doesn't make sense bad writing.

Ln98-100; TBS balloons are already being used for this purpose, this is not new…..

<Reff>; take out <> if needed use an overbar.

Section 2; WEATHER CONDITIONS; provide overall weather conditions and a table for cases here.

3. INSTRUMENTS

3.1 OPC

3.2 OLV

4. METHOD

4.1 microphysical parameters

4.2 Data reduction

Averaged for 10 mins intervals…….and analysis

5. RESULTS

5.1 light fog case

5.2 heavy fog case

5.3 Discussions e.g. issues, problems

6. CONCLUSIONS

Bullet 1:OPC has issues ….

Bullet 2 : OPC underestimates Nd and LWC

Bullet 3  OVL did not measures <5 micron etc
 bullet 4 why OPC measurements wrong? Why OVL taken as ref?

-viewing issues?

-sampling issues 1 sec versus 10 min and 1 hr comparisons?

Other issues:

-table 3; why upper limit is different? This affects LWC

Fig 3; OLV has 3 x more Nd? Why is that?

Eq 5; take out mean brackets, no need

Section 4.1; move after introduction

Line 279; detection… sentence, take it out.

Fig07; after corrections OPC measure more than OLV???? But one of previous figs was different.

Ln306; not clear….. why has measurements more than at 20 micron? Data available but company did not say?

Ln320; check this parag based on previous graphs.

---

## Author Comment (AC2)

Q1: Was any additional housing applied for the OPC to protect it against humidity or rain? Was it lying down on the roof of the building, or on any platform above the roof? If yes how much height above the roof surface? Can you present any photos of the devices set up?

A1: At the rooftop of the Faculty of Physics is located Radiation Transfer Laboratory where are conducted measurements of the optical and microphysical properties of atmospheric aerosols and clouds, as well as components of radiation fluxes and sensible and latent heat fluxes at the Earth's surface. Some devices, as Oxford Laser shadowgraph, are mounted just for a period of time, in this case for two months as the shadowgraph has a waterproof case. The OPC-N3 was mounted next to it just in the case of high probability of fog events without any protection. The picture of both devices mounted at Radiation Transfer Laboratory will be added.

Q2: In table 2, in the text it is written that OPC sampling was 10 s, then averaged up to 1 minute, in the table it is 1 minute sampling time, please make it consistent.

A2: Yes, I will correct in the table.

Q3: Equation 3, please check if all variables are explained, what is  $pix^2$ , is i here another variable or just index?

A3: Yes, We will add missing information.

Q4: Why there was double averaging applied? Why not straight average from 10s to 10 minutes? Please elaborate on how it would change if you would calculate it from 10 s, which was as far as I understood, basic sampling time.

A4: There is no difference in making an average first to 10s and next to 10 minutes, the standard deviation does not change. For clearer reading, it will be corrected that the average will be done right away to 10 minutes.

Q5: How uncertainty would change if you also consider Poisson statistics which represents a random error in the measurements?

A5: We have performed measurements to consider how big the impact will have on uncertainty the Poisson statistics. In Fig. A and Fig B are shown the values of errors for OPC-N3 and for ShadowGraph. The x axis in both figures represents the value for which the uncertainty was calculated and the y axis the uncertainty value. The plot represents how big the impact on total uncertainty has Poisson statistics. Error derived from Poisson statistics gives a greater contribution to the overall uncertainty for cases where a small number of droplets were counted. In the revised manuscript, the contribution to the overall uncertainty from Poisson statistics will be taken into account.

Q6:What was the reason to do the 1 hour averaging? Why not 0.5h or 2 hours? It should be elaborated, how was it representative?

Figure 1: The figure shows the dependence of the error on the measured value from OPC-N3. Colors mark the contribution of the error from the Poisson statistics (blue) and the measurement uncertainty (orange), the total measurement error is shown in yellow. In Fig. a) the uncertainty was calculated for LWC on b) number concentration.

Figure 2: The figure shows the dependence of the error on the measured value from ShadowGraph. Colors mark the contribution of the error from the Poisson statistics (blue) and the measurement uncertainty (orange), the total measurement error is shown in yellow. In Fig. a) the uncertainty was calculated for LWC on b) number concentration.

A6: ShadowGraph in one hour sample XX of air. One run of Shadowgraph was 10 minutes. Making plots every hour allows for receiving a smooth droplet size distribution spectrum from this device. The minimum averaging time was chosen with a smooth spectrum to observe the dynamics of fog.

**Q7: Is it 2:00:59 - 3:01:07 really an hour or a little bit more? I understand it is a minor issue, but it just looks strange.**

A7: The ShadowGraph collects the data in intervals of 10 minutes, between one run and another, there is a small brake 1-2 seconds for writing the files. As there is a small interval between runs, therefore, the two runes from the first ten minutes of the instrument's operation at the hour do not fall out equally in time. Each plot was made by averaging data from 6 runs of ShadowGraph which gives one hour.

Q8: How it differs from other periods? Can authors present the temporal evolution of droplet size distribution for all sampling periods (at least in the appendix)? The authors should explain to the readers why the analyzed period and later case study in section 4.1 was better than the rest of the time series.

A8: The case of fog from 16-17 November 2020 was chosen for a longer description as it was was the longest period of fog event registered during this study. Other cases of duration between 50 minutes up to 230 minutes. As ShadowGraph samples a small amount of air, making a smooth DSD spectrum is possible from the data aggregated in one hour. For other cases of fog occurrence, it would give 1-3 plots, which wouldn't allow to show the evolution of the fog case.

Q9: Authors with good results applied the Refractive Index correction. Please elucidate if all data presented are based on RIOPC or RIwater because it is not clear to me. Can you present any figure on how the correction influenced the measurements (at least in the appendix)?

A9: During the study we have checked if making a correction of Refractive Index to the data obtained from OPC-N3 would improve the results. The obtained results are inconclusive if the correction improves the data. The RI correction shifts the droplets measured by OPC-N3 to higher values, this improves the LWC comparison. However, the spectra of ShadowGraph and OPC-N3 are less compatible. In the manual, it is not well described how OPC-N3 converts the light scattering to droplet radii. The procedure can be not straight forward Mie Theory. Therefore, applying the correction does not improve the data so well. In the whole article, the standard (assumed by the manufacturer  $RI_{OPC}$  was used). We can add in the appendix the same analysis for  $RI_{water}$ .

Q10: Is it possible to apply any correction function for all factors influencing the OPC measurements (internal temperature, humidity, refractive index)??

A10: Internal temperature and humidity are not just shifted in comparison to ambient values, those factors are influenced, for example, by sun heating of the device. It can be seen from Figure 2 that internal temperature and humidity had a rapid change after sunrise. The refractive index correction can be explained in the appendix.

---

## Author Comment (AC3)

**Q1: General**

A1: We suggest in the manuscript the usage of OPC-N3 as a drone device for fog measurements. However, as the main purpose of OPC-N3 is for PM measurements, our goal of this article was to show that OPC-N3 can be used to detect fog particles. This article focuses on the comparison of OPC-N3 with OLV. We have used OPC-N3 for making vertical profiles of fog, however, before it's publication we wanted to make the first article showing that OPC-N3 is possible to detect fog droplets.

**Q2: Language and citations**

A2: The referee suggested some changes of the sentence structure and additional citations which will be done. Additionally, the text will be sent for language correction.

**Q3: Article structure**

A3: The referee suggested to change the article structure. The section "Instruments and methods can be divided into two separate sections. The proposition of the referee was to discuss the results as a case of light fog and heavy fog case. The article focuses on the calibration of OPC-N3 in all cases of fog measurements, and the calibration is done with the reference device OLV. During the measurements, only one case which is described in the manuscript, was longer than two hours and allowed for case analysing. Therefore, we would like to maintain the current structure of this paragraph. The conclusions can be rewritten in the form of bullets for better reading.

**Q4: Why OPC-N3 measurements wrong? Why OLV is seeing 3 times more Nc than OPC-N3?**

A4: OPC-N3 is a cheap optical counter. The manufacturer does not provide clear information about: - processing of the data (how light scattering is changed to the radius of droplets); - how OPC-N3 is built inside; - how is measured, the sampling volume, and how this is affected by the speed of the fan.

Without that information, it is hard to determine why OPC-N3 is detecting fewer droplets in comparison with OLV. We have come up with several possible scenarios in which the observations may be underestimated. Processing of the data - for example, assuming one RI for all particles - can lead to wrong droplet assignment to specific radii leading to lowering LWC. The fan speed forces the flow in OPC-N3 and changes in time. Therefore, the data are corrected by the manufacturer to take this effect into account, however there may be some bias which leads to a systematic lowering of the number concentration of droplets. The air is sucked into the OPC-N3 through a narrow inlet, inside we suppose the flow has no special path and expands throughout the whole device. This can affect the concentration of droplets in OPC-N3. Electronics inside OPC-N3 heats the surrounding which can lead to the arise of temperature and lowering humidity (see Section 2.4.2 and Fig. 2) and result in evaporation of droplets.

**Q5: Why OVL taken as reference?**

A5: OVL is a high quality device which provides particle and droplet sizing measurements in real-time. It was used for droplet characterisation in clouds giving good results. OVL is waterproof, which allows for installation in our rooftop laboratory.

**Q6: Why in table 3 the upper limit is different? Why has measurements more than 20 micron? Data available but company did not say?**

A6: The manufacturer of OPC-N3 provides information that the upper limit of the last bin is 20 microns. Big droplets of radius 19 microns are rare, however, when the number of droplets is multiplied by its density, it can be seen that the spectrum of mass has an abrupt peak in the last bin of OPC-N3 (Figure 1 upper panel). By consulting with Alphasense, it was obtained information that indeed the last bin can count also bigger droplets, however, as usually OPC-N3 is used for PM calculations and such big aerosols are not often, for PM calculation the assumption of upper limit as 20 micros is sufficient. Therefore, the upper limit of the last bin of OPC-N3 was chosen arbitrary and it can measure droplets higher than 20 microns. That is why in Section 2.4.1 (Scope of compliance between OPC-N3 and ShadowGraph), based on data from ShadowGraph, we estimated that the last bin of OPC-N3 can calculate the droplets up to 25.02 microns.

**Q7: Fig. 07: Why OPC-N3 measures more LWC than OLV**

A7: The OPC-N3 vDSD is smaller than from OLV. The plots consist of two scales, the scale for OPC-N3 is on the right side - orange, and is in a lower range than the scale for OLV (left, blue scale).

---

## Referee Report (RR1)

Nurowska et al

Feb 09 2023

Applicability of low cost……

Decision: Minor revisions

General

This work compares OPCn3 and OL VisiSize D30 observations for fog conditions, and evaluate uncertainties. The found out that corrections on OPC Nd is needed because of Ref Ind and measured size ranges. OPC Nd is found to measure Nd less than Nd from VisiSize. OPC T and RH are also found to be different compared to met obs. They also stated that OPC can be used for fog measurements.

Minor issues:

Take out UAV from the abstract

Ln 75; that needs a ref

Eq 1 and 2; sigma is the extinction coeff

DSD(r) show it as N(r)

181-182; why you need these lines….

Eq. 5; Vind,I; no need for "ind":

vDSD(r) change it to V(r)

Eq. 2; how come N(r) density is small compared to Visi N(r) but V(r) density is larger than V(r), please check it

Why you think that Visi measurements correct?

Fig 4; Visi has 2xLWC compared to OPC but same reff; how that is possible?

Fig 5; how do you get sd? Based on man values or raw data?

Fig 6; how did you get these means define them

Fig7; why you used a lower threshold of 9.73 micron? Why don't plot also Nd and LWC time series?

Fig A3; after correction results is worst than before, then why you do it?

---

## Referee Report (RR2)

Corrections on Nurowska et al

Reviewer B

03/23/2023

**Major corrections**

Table 2; what are the units in the table, not clear.

Sampling of what?

Eq 6;  ……are given as, respectively, as…

Eq. 9 is wrong.

Fig. 4; which day? Provide clearly this info in figs.

Fig 5 and 6; where the data comes from? Case study? Or segment what?

CONCLUSIONS;

after first sentence say: The following results are obtained from the current work. Or similar.

Ln350; new N3 detect particle up to 40 micron, clarify this here.

AE-51 device on flying platforms??? This is not validated and not possible likely due to calibration issues. Please take it out.

Table A1; units please.

Ln420;  double however???? Take out one.

Fig A3; After correction it becomes worst (red ones); why we should use the corrected one? If corrected one is correct, then ShadowG results are not reliable. Please clarify to me.

Table 1; gm-3 is in wrong locations. Also, what is the equation for these coeff. It is not clear to me.

---

## Author Response (AR2)

**1 Report 2,**
**Anonymous Referee 1, report 10 Feb 2023**

**Q1:Take out UAV from the abstract**
**A1:** We would like to leave the sentence about UAV as it is, because it is showing our motivation for this and future studies. In this article we are not showing the results from the UAV, however, we already have been using OPC-N3 for UAV measurements. This is the first article and we are willing to publish in the future a continuation including the results from drones.

**Q2: Ln 75; that needs a ref**
**A2:** I suppose this is referring to the 76 line "Optical devices, such as the lidar or the celiometer, cannot penetrate thick fog to retrieve information about their vertical structure". We added a reference.

**Q3: Eq 1 and 2; sigma is the extinction coeff**
**A3:** Yes, it was a mistake, fixed.

**Q4: DSD(r) show it as N(r). vDSD(r) change it to V(r).**
**A4:** We use DSD(r) and vDSD(r) as it is used in other papers [1, 2]. Formulas for DSD(r) and vDSD(r) are the following:

$$DSD(r_b) = N_b \cdot (V_b \cdot \Delta r_b)^{-1}, \tag{1}$$

$$vDSD(r_b) = DSD(r_b) \cdot r_b^3, \tag{2}$$

where $N_b$ number of droplets in a bin, $V_b$ - volume of a bin, $\Delta r_b$ width of the bin, $r_b$ mean bin droplet radius.

**Q5: 181-182; why you need these lines....**
**A5:** Lines 180 till 185 describes the calculation of volume in ShadowGraph device. The sampled volume depends on the droplet size. The volume sampled is required i.e. for number concentration calculation.

**Q6: Eq. 5; Vind,I; no need for "ind":**
**A6:** Fixed.

**Q7: Eq. 2; how come N(r) density is small compared to Visi N(r) but V(r) density is larger than V(r), please check it**
**A7:** I suppose this question was about Fig.2, not Eq.2. I suppose the referee means DSD by N(r) and vDSD by V(r), which is confusing. Fig.2 shows the droplet size distribution DSD(r) (number concentration divided by the bin width) compared with the volume size distribution vDSD(r) (DSD multiplied by the mean radius of the bin to the third power). Apart from the last bin of OPC-N3, the DSD and vDSD from ShadowGraph are bigger than from OPC-N3

(the figure has different scales on the left and on the right axes). Fig.2 shows that presenting the data in the form of vDSD allows for better emphasizing the role of bigger droplet bins in the contribution to the LWC.

**Q8: Why you think that Visi measurements correct?**
**A8:** There are papers (Nowak et. al. 2021, Mohammadi et. al. 2022) validating VisiSize for atmospheric microphysic measurements of droplets (i.e., in cloud / fog).

**Q9: Fig 4; Visi has 2xLWC compared to OPC but same reff; how that is possible?**
**A9:**LWC is the amount of total liquid water. The $r_{eff}$ of both devices is similar, however, the ViSize D30 is registering more droplets than OPC-N3, that is, why the LWC in ViSize is bigger than in OPC-N3.

**Q10: Fig 5; how do you get sd? Based on man values or raw data?**
**A10:** I do not understand what the referee meant by "sd". The Fig.5 show effective radius. Formula for $r_{eff}$ is given by Eq. (6).

**Q11: Fig 6; how did you get these means define them**
**A11:** The following text was added to the manuscript. To calculate the mean radius $r$ the following formula was used:

$$r = \left( \sum_{i=1}^{m} r_i \cdot N_i \right) \cdot \left( \sum_{i=1}^{m} N_i \right)^{-1}, \tag{3}$$

To calculate the mean surface radius $r_S$ the following formula was used:

$$r_S = \left( \left( \sum_{i=1}^{m} r_i^2 \cdot N_i \right) \cdot \left( \sum_{i=1}^{m} N_i \right)^{-1} \right)^{1/2}, \tag{4}$$

To calculate the mean volume radius $r_V$ the following formula was used:

$$r_V = \left( \left( \sum_{i=1}^{m} r_i^3 \cdot N_i \right) \cdot \left( \sum_{i=1}^{m} N_i \right)^{-1} \right)^{1/3}, \tag{5}$$

**Q12: Fig7; why you used a lower threshold of 9.73 micron? Why don't plot also Nd and LWC time series?**
**A12:** The Fig. 7 c) presents the LWC time series for cases of automatic fog detection. The process of automatic fog detection was described in Section 3.1. I will change the sentence 9.73 for a reference to Sec. 3.1, it will be more clear.

**Q13: Fig A3; after correction results is worst than before, then why you do it?**

**A13:** There is a difference in data between OPC-N3 and ShadowGraph. We tested if this inconsistency is due to RI assumed in OPC-N3. In a previous review, we were asked to add information on how the correction for the refractive index was carried out in the Appendix. The results of RI correction are discussed in the last point of Section 6. "Conclusion". Testing if RI correction is working provides information that probably Alphasense has a more sophisticated (than only based on Mie theory) built-in processing of the data while assigning it to the bin size.

**References**

[1] Zhouhang Li et al. "Effect of liquid viscosity on atomization in an internal-mixing twin-fluid atomizer". In: *Fuel* 103 (2013), pp. 486–494. ISSN: 0016-2361. DOI: 10.1016/j.fuel.2012.06.097.

[2] M. Mohammadi et al. "Cloud microphysical measurements at a mountain observatory: comparison between shadowgraph imaging and phase Doppler interferometry". In: *Atmospheric Measurement Techniques* 15.4 (2022), pp. 965–985. DOI: 10.5194/amt-15-965-2022.

---

## Author Response (AR3)

**1 Report 2, March 2023**

**Q1:Table 2; Sampling**
**A1:** Added units consistently. Data were collected by ShadowGraph in runs, one run was of 10 minutes. During one run there was collected an information about each observed droplet passing through field of view of ShadowGraph.

**Q2:Eq. 6 and 9**
**A2:** Corrected.

**Q3:Fig. 4; Fig 5 and 6; which day? Provide clearly this info in figs**
**A3:** At the beginning of Section "Microphysical properties of fog" was added information that the graphs were made collectively for all the dates when the fog was registered.

**Q4:after first sentence say:**
**A4:** Added.

**Q5:new N3 detect particle up to 40 micron, clarify this here**
**A5:** Added.

**Q6: AE-51 device on flying platforms??? [...] Please take it out**
**A6:** Deleted

**Q7:Table A1; units please**
**A7:** Added in more clearly way.

**Q8: Ln420; double however????**
**A8:** Removed.

**Q9: Fig A3; After correction it becomes worst (red ones);**
**A9:** We have seen that for example LWC from OPC-N3 is lower than from ShadowGraph. We were analysing what could be the reason of that. We wanted to check if the underestimation may have been due to different RI assumption in OPC-N3. We have performed analysis which is in appendix showing how the looks the data from OPC-N3 after the correction of RI index. As a result indeed the correction increased the LWC, it was due to shifting the droplets to higher bin. However the correction worsened the compliance of mean droplet diameter between OPC-N3 and ShadowGraph. We conclude that the OPC-N3 software attributing droplets radii does not rely directly on Mie Theory. Secondly we discuss other reasons why OPC-N3 can be lowering the values like LWC.

**Q10: Table 1; gm-3 is in wrong locations. Also, what is the equation for these coeff. It is not clear to me**
**A10:** The linear regression coefficients, Pearson's correlation coefficient (PCC) and the root mean square error (RMSE) for LWC, $N_c$, $r_{eff}$, $r$, $r_s$, $r_V$. The linear

regression was obtained for data without RI correction and with RI correction applied. The linear regression curve of the form y = ax + b was fitted to the respective quantity, where x is the measured value from OPC-N3, a and b are calculated coefficients and y is the measured value from ShadowGraph.